# Effect of Biochar from Oat Hulls on the Physical Properties of Asphalt Binder

**DOI:** 10.3390/ma15197000

**Published:** 2022-10-09

**Authors:** Camila Martínez-Toledo, Gonzalo Valdés-Vidal, Alejandra Calabi-Floody, María Eugenia González, Oscar Reyes-Ortiz

**Affiliations:** 1Department of Civil Engineering, University of La Frontera, Temuco 4811230, Chile; 2Department of Chemical Engineering, University of La Frontera, Temuco 4811230, Chile; 3Department of Civil Engineering, Military University of Nueva Granada, Bogotá 111711, Colombia

**Keywords:** biochar, asphalt binder, physical properties

## Abstract

The purpose of this study was to verify the feasibility of using biochar from oat hulls (BO) as a potential bio-modifier to improve the physical properties of conventional asphalt binder. The BO and asphalt binder were characterized by confocal (fluorescence) laser microscopy, scanning electron microscopy and Fourier transform infrared spectroscopy. Then, an asphalt binder modification procedure was established and modifications with 2.5, 5.0 and 7.5% of BO on the weight of the asphalt binder were evaluated, using a particle size < 75 µm. The physical properties of the evaluated modified asphalt binder with BO were: rotational viscosity in original and aged state, aging index, Fraass breaking point, softening point, penetration, penetration rate and storage stability. The results indicated that the BO has a porous structure, able to interact with the asphalt binder by C=O and C=C bonds. In addition, modification of the asphalt binder with BO increases the rotational viscosity related to high-temperature rutting resistance. The results obtained from the Fraass breaking point and softening point indicated that the use of BO extends the viscoelastic range of the asphalt binder. In addition, the evaluated modifications present low susceptibility to aging and good storage stability.

## 1. Introduction

Nowadays, material engineers and scientists are becoming increasingly interested in modifying the properties of conventional asphalt binders using waste from an industry that causes tremendous pollution. This interest is based mainly on increasing the service life of asphalt pavements, generating a lighter environmental impact [1].

The performance of asphalt binder under different climatic and traffic conditions causes the appearance of certain failure modes that affect the structure of the pavement. One of the important characteristics in the performance of asphalt binder is its work range, which is the ability to change from a solid-elastic to a viscous state (viscoelastic range) according to the temperature changes it undergoes [2]. This, added to its response to the application of loads at different speeds, conditions the performance or rheological behavior of the asphalt binder, which affects the performance in a pavement structure [3]. 

One of the alternatives used to achieve more resistant and durable pavements is the use of different types of modified asphalt binder, which can improve such properties as rutting resistance, fatigue cracking, thermal cracking and other types of deterioration. In this context, the use of commercial polymer has contributed to the performance of asphalt binder, with SBS (styrene-butadiene-styrene) being one of the most used polymers in industry for the modification of asphalt binder. However, its use also involves a considerable increase in the cost of the end product compared to a conventional asphalt binder [4]. Therefore, various studies have been conducted of late that seek to assess other material additives that can improve the rheological properties of the asphalt binder. One of the lines of study is the addition of nano-modifiers, such as carbon nanotubes, graphene oxide, nanosilica and others [5,6,7,8]. Although these nano-modifiers have yielded good results, their performance and production costs mean that their industrial use is not yet economically viable as an asphalt binder-modifying additive [9,10]. On the other hand, micro-sized modifying materials (<150 µm) or micro-materials have been used with good results to improve the performance-related properties of the asphalt binder. Such is the case of asphaltite, graphite and activated carbon [11,12,13]. 

In this same line of enquiry, an alternative potential to modify asphalt binder may be biochar. Biochar is a solid, carbon-rich by-product obtained from pyrolysis [14]. Pyrolysis is a thermochemical process carried out between 300 and 1000 °C in the absence of oxygen to produce biofuels and transform waste biomass into biochar [15,16]. This technique reduces emissions caused by biomass incineration and reduces the use of landfills for waste disposal [17]. In addition, the biochar produced can be used as a tool for reducing greenhouse effect gases (GEG) and sequestering CO_2_ in the air [18,19,20].

Research into the use of biochar from different types of biomass of plant origin, with particle sizes < 75 µm to modify asphalt binder, have shown that it can increase the anti-aging properties, maintain resistance to thermal cracking and improve rutting resistance [1,14,21,22]. In addition, it could be a cost-competitive modifier in relation to other modifiers with similar characteristics, such as graphite and activated carbon [23,24,25,26]. However, there is a very limited number of studies on the effects of biochar in asphalt binder [27], and studies of biochar from oat hulls in their application as an asphalt binder modifier do not exist. Biochar from oat hulls has a high content of carboxylic groups, low heavy metal content and high carbon content [28,29]. Additionally, some countries have states or regions that have oats as a large surface of their agro-industrial crops. For example, the region of La Araucanía in Chile has the widest distribution of its crops with oats, of which 62.1% of the surface is dedicated to agriculture [30]. These crops generate a large amount of residual biomass which mainly comes from oat hulls. Oats are estimated to have a hull:grain ratio being 30:70% by weight [31].

Accordingly, the aim of the present study is to verify the feasibility of the use of biochar from oat hulls (BO) obtained by slow pyrolysis at 300 °C as a potential bio-modifier to improve the physical properties of asphalt binder.

## 2. Materials and Methods

### 2.1. Materials

The asphalt binder used in this study is a conventional CA-24 asphalt binder (according to Chilean specifications), and its characterization is in Table 1. 

The biochar selected is a by-product of the slow pyrolysis of oat hulls for 2 h of residence time, with a heating rate of 3.6 °C/min and a pyrolysis treatment temperature (PTT) of 300 °C. Figure 1 shows the schematic diagram of the pyrolysis used, where the reactor is stainless steel and has a capacity of 0.117 m^3^, approximately. This reactor is equipped with an inert gas connection (N_2_) with a flow of 0.001 m^3^/min to purge the oxygen generated during the reaction. The furnace parameters were programmed using a programmable logic controller (PLC). The oven is heated by the resistances installed inside, the temperature of which is controlled by a K-type thermocouple sensor (Ni-Cr-Ni). According to the procedure, the volatile by-products present in the synthesis gas were condensed by the circulation of water at room temperature; the reception of these condensable gases (bio-oil) took place in a TAR container, while the gaseous fraction was released into the atmosphere. 

When the reaction was complete, a 44% BO yield was obtained. The BO obtained was subjected to a mid-sized reduction process using a 180 W grinder with a processing time of 30 s. Then, to obtain a particle size <75 µm, a sieve with a 0.075 mm mesh (N° 200) was used. Figure 2 shows the BO production sequence.

### 2.2. Asphalt Binder Modification Procedure

To establish the procedure of asphalt binder modification with BO, a matrix was created, composed of: three mixing times (30, 60, 120 min), three modification temperatures (160, 170, 180 °C) and a single BO content, equivalent to 5.0% in weight of the asphalt binder. Modification temperatures higher than the mixing temperature of the asphalt binder were considered because the addition of BO can increase the viscosity of asphalt. For each configuration, the modification of 1000 g of asphalt binder was carried out. Initially, the asphalt binder was heated in an oven to 130 °C for 30 min and then BO was added and the mixing process began, which was carried out using an electronic stirrer at 350 rpm. The BO was added gradually. The modification temperature was verified every 5 min using a digital thermometer with an accuracy to 0.1 °C until the mixing time was complete. Then, the samples were analyzed by confocal laser microscopy to evaluate the distribution and integration of the BO with the asphalt binder. The results determined that, with a constant temperature of 160 °C and 30 min of mixing, a homogenous sample between the BO and the asphalt binder is obtained without clumping. 

### 2.3. Experimental Plan

For the temperature and selected mixing time, the additions of 2.5%, 5.0% and 7.5% of BO (<75 µm) on the weight of the asphalt binder were evaluated, along with a reference sample of asphalt binder (CA-24). Figure 3 provides the experimental plan to assess the effect of modification with BO on the physical properties of the asphalt binder at different stages. This experimental plan used physicochemical characterization procedures from scanning electron microscopy (SEM), energy-dispersive x-ray spectroscopy (EDS), confocal (fluorescence) laser microscopy, infrared spectroscopy (FTIR) and assay methods to evaluate physical properties of the asphalt binder related to its mechanical behavior and industrial use.

### 2.4. Test Methods

#### 2.4.1. Physicochemical Characterization of Raw Materials and Samples

The SEM and EDS assays made possible an advanced analysis of the surface of the materials included in this study. In this way, the microscopic morphology, particle size and elemental chemical composition of the surface of the materials were obtained using a SU3500 scanning electron microscope from Hitachi High-Technologies Corporation, Tokyo, Japan, equipped with an energy-dispersive X-ray detector.

To verify the distribution and integration of the BO in the asphalt binder after the modification process, a FV1000 Olympus confocal laser (fluorescence) microscope was used in the excitation/emission spectra in 2 wavelengths: 488/590 nm and 530/590 nm, with a 20× magnification and a 30 µm depth in steps of 2 µm/slide.

To detect the functional groups present on the surface of the carbon materials, FTIR was carried out using Perkin Elmer Spectrum Two infrared spectrometer that includes an ATR (attenuated total reflectance) system. The FTIR assay was executed between 3400 cm^−1^ and 400 cm^−1^, with a resolution of 4 cm^−1^ and a laser repetition rate of 20 at 20 °C.

#### 2.4.2. Evaluation of the Physical Properties of Modified Asphalt

The rotational viscosity test (RV) was applied to measure the flow resistance and the workability of the samples at high operating temperatures of 52, 58, 60, 64, 70 and 76 °C, and at working temperatures of 135 and 165 °C, in both original and short-term aged samples using a rolling thin-film oven. The testing procedure was done according to AASHTO T316-19 using a Brookfield RVDV-III ULTRA rotational viscometer at a shear speed of 20 rpm.

The rolling thin-film oven test (RTFOT) was performed to simulate the short-term aging that occurs in asphalt during the manufacturing process of the mixture and its spreading in the pavement, subjecting the samples to the effect of heat and air at 163 °C for 85 min according to the procedure described in AASHTO T240-13 using a Controls Group 81-PV1612 thin-film rolling oven.

The aging index (I_ag_^r^) is a parameter used to determine the susceptibility to short-term aging of the asphalts evaluated, using Equation (1):(1)$$I_{{\mathrm{ag}}^{r}{=V}_{\mathrm{RTFOT}}{/V}_{\mathrm{original}}}$$
where,
I_ag_^r^ = aging rate of the asphalt binder,RV_RTFOT_ = rotational viscosity of the asphalt binder aged by RTFOT (poises),V_original_ = rotational viscosity of the asphalt binder in original state (poises).

The Fraass breaking point test was performed to evaluate the behavior of the asphalt at low operating temperatures since it determines the transition temperature at which asphalt goes from a viscoelastic state to an elastic state [33]. The testing procedure was done according to EN 12593:2007 using the Control Breaking Point apparatus, subjecting a thin film of asphalt to successive bending cycles at decreasing temperatures at a cooling rate of 1 °C/min. 

The softening point test (SP) was conducted to determine the softening point temperature of asphalts evaluated according to the procedure described in ASTM D36-76 using the ring and ball apparatus. The softening point temperature obtained defines the transition of the asphalt from a viscoelastic state to a purely viscous state [34]. The lowest temperature at which the asphalt sample, suspended in a horizontal ring, was forced to fall due to the weight of a steel ball while it was heated at 5 °C/min in a water bath was recorded.

The penetration test (Pen) described in ASTM D5-13 was used to determine the hardness of the asphalts according to the penetration depth of a stainless-steel needle on their surface, the greatest values of which indicate softer consistencies of the material and vice versa. The procedure was done using a B057 automatic penetrometer, applying 100 g of weight for 5 s at 25 °C.

The penetration index (PI) or Pfeiffer index is a parameter to describe the thermal susceptibility of asphalt to temperature changes. In addition, it offers an indication of its colloidal structure and rheological behavior [32,33]. Table 2 provides a description of the characteristics of the asphalt binders based on the PI values. 

This parameter can be calculated using Equation (2), with the results of the penetration tests at 25 °C and softening point.
(2)$$\mathrm{PI}=\frac{1952\;-\;500\cdot \log\left(\mathrm{Pen}\right)\;-\;20\mathrm{SP}}{50\cdot \log\left(\mathrm{Pen}\right)\;-\;\mathrm{SP}\;-\;120}$$
where,
PI = penetration index of the asphalt binder, Pen = result of the penetration test at 25 °C,SP = result of the softening point test using the ring and ball apparatus.

The storage stability test was conducted according to the procedure described in ASTM D58-92. Although this assay evaluates the storage stability of asphalts modified with polymers, like SBS, it was also used to evaluate the asphalt modification with BO because the test results offer the data required for the analysis of this study. The modified asphalt binder samples were poured into 1” diameter tubes and conditioned in an oven at 163 ± 5 °C for 48 h. The samples were then conditioned in a freezer at −6.7 ± 5 °C for 4 h. Next, the tubes were cut into three sections of equal length, and the central part of each tube was discarded. Finally, these samples were poured into appropriately marked rings to perform the softening point test by means of the ring and ball apparatus according to ASTM D36-76. To consider storage stability good, the difference in the results of the softening point between the upper and lower sections of the sample cannot exceed 5 °C [33].

## 3. Results

### 3.1. Physicochemical Characterization of Raw Materials and Samples

#### 3.1.1. Scanning Electron Microscopy (Sem) and Energy Dispersive X-ray Spectroscopy (EDS)

The image of oat hulls in Figure 4a obtained by SEM shows a great predominance of large particles (≤1 mm) with heterogenous geometry, whereas the micrograph of the BO in Figure 4b shows particles of different sizes with an irregular surface and a certain pore development, with few cases of vitreous surfaces.

Table 3 shows the results of SEM+EDS micro elemental analysis of the evaluated samples. It is observed that the predominant chemical element is carbon. Sulfur as well as oxygen are elements in common among all BO-modified asphalt binder samples. Thus, Fisher’s least significant difference (LSD) method with a 95% confidence level was used for multiple comparisons of sample means. It was determined that there is a significant difference between the amount of carbon (C) in the reference asphalt binder (CA-24) and the BO-modified asphalt binders. However, among the modified asphalt binders, there is no significant difference in the amount of this chemical element (C), despite increasing the percentage of BO in the sample. With regard to the amount of sulfur (S) and oxygen (O), there are no significant differences between the samples of asphalt binder modified with BO, but the latter element (O) decreases significantly compared to the modifier (BO). It is also possible to observe other chemical elements in the samples modified with BO, such as silicon, potassium and phosphorus. These are inherent to the modifier used (BO). The higher the percentage of BO, the higher the presence of these elements.

#### 3.1.2. Confocal (Fluorescence) Laser Microscopy

The images in Figure 5 show that for all the percentages of modification with BO a good distribution in the asphalt binder is achieved, without the presence of clusters. This indicates that there is a homogenous integration between the two materials, which is why the modification procedure used is the one suitable for the asphalt binder modification with up to 7.5% BO. On the other hand, it is observed that the maximum sizes of BO particles are around 30 µm despite having separated the BO fraction using a 75 µm sieve. This result is attributed to the downsizing process used, because it was highly efficient in the production of small particles due to the characteristics of the grinder and the BO processing time.

#### 3.1.3. Fourier Transform Infrared Spectroscopy (FTIR)

Figure 6 shows the spectra of the BO, CA-24 and the asphalt binder modified with BO. In the BO, the main infrared signals are in the band 3005.0 cm^−1^ corresponding to C-H, 1540.5 cm^−1^ corresponding to C=C type rings and 1077.61 cm^−1^ corresponding to C=C or C-O-C bonds. These peaks can be attributed to the lignin, cellulose and hemicellulose of the biochar used (BO), which have a content of 7.5%, 34.3% and 26.0% respectively [28,29]. The spectrum of CA-24 shows more noise than that of other studies but has characteristic signals similar to them [35]. The stretches at 2919.38 cm^−1^ and 2850.99 cm^−1^ correspond to methylene CH_2_, the peak at 1591.9 cm^−1^ corresponds to the C=C rings of benzene, the peak at 1454.51 cm^−1^ corresponds to CH_2_, the peak at 1375.33 cm^−1^ corresponds to methyl CH_3_ and the peak at 1023.8 cm^−1^ corresponds to C-O-C or C-O. The peaks at 810.18 cm^−1^ and 723.56 cm^−1^ correspond to C-H. Several peaks are noted in the spectra of the modifications that corroborate the interaction between the functional groups present on the surface of the BO and the asphalt binder. The main interactions observed are C=O and C=C bonds in the band of 1021 cm^−1^ (represented by a vertical dotted line). When the percentage of BO in the asphalt binder is increased, a trend reduction in the transmittance is observed, confirming the interaction between the two matrices, however more trials are needed to quantify this reduction. The CA-BO2.5 and CA-BO5.0 samples present equal intensity of interaction. By contrast, the CA-BO7.5 sample is the one with the greatest interaction between the polymeric mixture, observing a greater reduction in the transmittance. In addition, the modifications of the asphalt binder bands suggest an interaction with the BO through functional groups present on the surface of the BO, forming C=O and C=C bonds without there being large differences in the fixations between the mixtures of 2.5%, 5.0% and 7.5% of BO.

### 3.2. Analysis of Physical Properties of Modified Asphalt

#### 3.2.1. Rotational Viscosity (RV)

The results shown in Figure 7a indicate that the RV of the asphalt binder modified with BO increased for all the addition percentages considered and temperatures evaluated in relation to the CA-24. For example, at 60 °C the viscosities increased by 27.5%, 32.1% and 47.1% for the CA-BO2.5, CA-BO5.0 and CA-BO7.5 samples, respectively. These results show that as the amount of BO in the asphalt binder increases, a growing trend is registered in the increase in RV. This behavior indicates that the modification with BO can increase the viscosity of the asphalt binder in a high range of operating temperatures, which could aid in improving rutting resistance of the wearing course of a pavement [36]. These results are consistent with other studies, such as the study by Muhammed et al. [37], who modified asphalt binder with biochar from the pyrolysis of ground walnut shells and apricot seed shell granules, determining that the increase in the concentration of these bio-modifiers also increased the rigidity of the asphalt binder. This effect also was identified by Walters et al. [38], who used a thermo-chemical process to convert pig dung into bio-oil using biochar obtained to modify asphalt binder. The results indicated that the viscosities tended to increase as the addition percentages of biochar in the asphalt binder increased. In relation to the particle size of BO used in the modification of the asphalt binder (<75 µm), the effect on the RV is considered high, agreeing with the report by Zhang, et al. in 2018 [21], who described the particles of BO < 75 µm as registering higher RV than those modifications with larger particles (between 75 and 150 µm). This effect is because a greater number of small particles can be located in the same surface area. In addition, the porous structure and surface area that the BO possesses can produce a better adhesion and interaction with the asphalt binder, reducing its fluidity and increasing its viscosity. In this same context, the structure of the BO would allow a possible absorption of the lightest components of the asphalt binder, thereby increasing its RV [35].

Figure 7b shows that the RV of the reference asphalt binder and the RV of the asphalt binder modified with BO increased compared to the results obtained from these same samples in their original state due to the oxidation process that the asphalt binders underwent after aging by RTFOT. It is also worth noting that the RV of the asphalt binder modified with BO is greater than the RV of the reference asphalt binder (CA-24). This increase in the RV is proportional to the amount of BO added to the asphalt binder. In this sense, at 60 °C the CA-BO2.5, CA-BO5.0 and CA-BO7.5 samples increased their RV by 28.4%, 42.1% and 57.6% compared to the RV of the CA-24, respectively. These increases are greater than those determined in the original state of the asphalt binder as shown in Figure 7a.

In Figure 8 it is noted that both the optimal temperature to obtain the recommended mixture viscosity (~2 poises) and the optimum temperature to obtain the compaction viscosity (~3 poises) of the asphalt binder modified with BO increased compared to the temperatures of the reference asphalt binder, at 5 °C and 7 °C respectively. The highest temperatures corresponded to CA-BO7.5 (164 and 158 °C) and the lowest temperatures to the reference asphalt binder (159 and 151 °C). Within this range are the optimum mixture and compaction temperatures of CA-BO2.5 and CA-BO5.0, respectively. According to the data shown in Figure 8, CA-BO5.0 increases 38% and 44% more with the RV at 135 °C than the biochar from ground apricot seed shell and ground walnut shell, respectively. By contrast, at 165 °C, the RV increases by 23% and 26% more than those same modifiers, respectively [37]. 

#### 3.2.2. Aging Index (I_ag_^r^)

Figure 9 shows the evolution of I_ag_^r^ for different evaluated temperatures. A trend is observed when the I_ag_^r^ decreases as the temperature increases. This indicates that the aging effect in the reference asphalt binder and the asphalt binder modified with BO decreases as the evaluated temperature increases. On the other hand, for the highest contents of BO in the modification (CA-BO5.0 and CA-BO7.5), greater values are obtained in the parameter I_ag_^r^, with increases that vary in the range of 5% and 8% compared to the reference asphalt binder. However, these fluctuations do not generate significant differences between the samples with 5.0% and 7.5% of BO and I_ag_^r^ of the reference asphalt binder (CA-24), consistent with the results from the Kruskal–Wallis test, where a significant value of 0.323 was obtained. With respect to the CA-BO2.5 sample, this presents a behavior similar to the reference asphalt binder, showing equal variances according to Levene’s test with a significant value >0.05, whereas the Student *t*-test results indicated that there is no significant difference between the means of CA-BO2.5 and the reference asphalt binder with a significant value equal to 0.680. These results could be due to the morphology of the BO particles, which are characterized by their heterogenous and porous shapes that could adsorb the asphalt binder, causing a reduction in the exposure of the asphalt binder to heat during the oxidation process, which would reduce the effects of aging [14,21]. On the other hand, the fluctuations obtained in the I_ag_^r^ parameter may be due to the loss of light compounds that the BO absorbed during the modification of the asphalt binder [35]. Nevertheless, the values obtained for the parameter I_ag_^r^ of the modifications evaluated in this study are lower than those specified in different standards, as in the case of the standard for Chile [32], which specifies for 60 °C a maximum I_ag_^r^ value of 4.43% higher than the maximum I_ag_^r^ value obtained at that temperature in this study.

#### 3.2.3. The Fraass Breaking Point vs. the Softening Point (SP)

The results show that the use of BO as an asphalt binder modifier can extend the viscoelastic range of the asphalt binder once the Fraass breaking point is reduced and the softening point is increased (Figure 10). In this respect, it is observed that the Fraass breaking point decreases as the BO content in the asphalt binder increases. For example, the CA-BO2.5, CA-BO5.0 and CA-BO7.5 samples achieved a breaking temperature of −8.5 °C, −9.5 °C and −10.0 °C respectively, being 2.0 °C, 3.0 °C and 3.5 °C lower than the breaking temperature of the CA-24. These results point to the asphalt binder modified with BO reducing the temperature at which the asphalt binder reaches the critical rigidity value after its fracture, and therefore its cracking [33]. On the other hand, it is observed that the softening point increases according to the increase in the amount of BO in the asphalt binder. For example, the CA-BO2.5, CA-BO5.0 and CA-BO7.5 samples reached a softening temperature of 51.0 °C, 53.8 °C and 55.3 °C respectively, increasing the softening temperature by 1.0 °C, 3.8 °C and 5.3 °C compared to the CA-24. The results show that the addition of BO increases the temperature range at which the asphalt binder can vary its behavior from a purely fragile state to a purely viscous state. Thus, the addition of this bio-modifier enables the asphalt binder to present a less fragile behavior at the same temperature as the reference asphalt binder. These results could justify an improvement in its response to cracking at low operating temperatures. By contrast, at the other extreme, they could show a greater capacity to resist higher temperatures, since its transition from a viscoelastic material to a viscous material occurs at a higher temperature, where the most common failures are rutting resistance. In that sense, an increase in the evaluated temperature ranges could mean an improvement in the resistance of the asphalt binder to the failures produced at low and high operating temperatures. According to the data in Figure 10, 5% of BO increases the SP of the asphalt binder by 8%, 12% and 16% more than the biochar DS-510F [39] and biochars from ground apricot seed shell and ground walnut shell, respectively [37]. Meanwhile, 7.5% of BO increases the SP of the asphalt binder by approximately 7% more than the biochar DS-510F [35,39].

#### 3.2.4. The Penetration (Pen)

The results provided in Figure 11 indicate that the addition of BO increases the hardness of the asphalt binder at 25 °C. The reduction in the penetration depth as the BO concentrations in the asphalt binder is increased is more obvious in the CA-BO5.0 and CA-BO7.5 samples, the results of which show similar values, with a decrease of 29.4% and 29.9%, respectively compared to the CA-24. However, the CA-BO2.5 presented a 16.1% reduction compared to the penetration of the CA-24. These results indicate that the asphalt binder modified with the different amounts of BO make it possible to increase the rigidity of the asphalt binder at the intermediate temperature, fulfilling the minimum penetration of 40 dmm by some standards that classify their asphalt binder by degree of viscosity, like the Chilean standard for example [32]. When using 5% or 7.5% of BO, the penetration of the asphalt binder decreases about 14% more than when using the same percentages of other modifiers, such as biochar DS-510F [35,39].

#### 3.2.5. Penetration Index (PI)

From the data provided in Table 4, the PI values of the asphalt binder modified with BO are similar to the CA-24 up to the content of 5.0%. For a BO addition of 7.5%, an increase in the PI is recorded, indicating a reduction in thermal susceptibility compared to the CA-24. For this BO content, the effect on the reduction of the Pen results was observed, and at the same time on the increase in the SP values, which made it possible to reduce thermal susceptibility. This effect on IP reduction was also observed in other studies where biochar from ground apricot seed shell and ground walnut shell was used as an asphalt binder modifier [37]. Additionally, the classification of all the asphalt binder modified with BO corresponds to the category of intermediate thermal susceptibility, understood as a penetration index of between −1 and +1 [32].

#### 3.2.6. Storage Stability

The results in Figure 12 indicate that the material corresponding to the upper superior of the sample records a lower SP than the lower section, which is observed for all the analyzed samples. With respect to the results obtained for both CA-BO2.5 and the CA-BO5.0, a difference is noted between the SP of the upper and lower sections of 1.5 °C and 2.5 °C, respectively. By contrast, the CA-BO7.5 sample shows a difference close to 4 °C. Differences in the SP in the range from 2 to 5 °C between the upper and lower sections of the sample are considered good storage stability [25]. In this sense, the results indicate that all the samples of asphalt binder modified with BO present good storage stability, demonstrating that the BO contents evaluated in the study would have good compatibility and interaction with asphalt binder, and that the modification process used enabled good bonding between phases. However, studies on asphalt binder modification report that to achieve a high homogeneity of the modified asphalt binder, the differences in the SP between the upper and lower sections of the sample should be ≤2.5 °C [40,41]. In that case, only the additions of 2.5% and 5.0% of BO fulfill this condition of high storage stability. 

Figure 13 is a summary of the effects of the BO on the physical properties of the asphalt binder.

## 4. Discussion

The present study was conducted to verify the feasibility of the use of biochar from oat hulls (BO) as a potential bio-modifier of the physical properties of conventional CA-24 asphalt binder. 

It is determined that the asphalt binder and BO interact positively due to C=O and C=C bonds of the functional groups present on the surface of both materials.It is shown that the BO can be distributed homogenously in the asphalt matrix, in all the addition percentages considered, without causing clusters.The rotational viscosity of the asphalt binder in the original and short-term aged states increased with the addition of BO. This increase was directly proportional to the amount of BO added to the asphalt binder.The resistance to aging of the asphalt binder was maintained with the addition of BO. The values obtained for the parameter of aging of the modifications evaluated were lower than the regulatory requirements.The viscoelastic range of the asphalt binder can be extended with the addition of BO, being proportional to the increase in the modifying content, and being able to reduce the thermal susceptibility of the asphalt binder.The use of BO increases the consistency at an intermediate temperature, reducing the penetration of the asphalt binder.The reduction of the penetration, the increase in the softening point and the increase in viscosity demonstrate that BO improves the performance-related properties of the asphalt binder at high temperatures.With up to 7.5% modification with BO, good storage stability of the asphalt binder is obtained.Future studies are suggested to assess the effect of different PTT and residence times on the properties of the BO as a modifying additive. In addition, the effect of a smaller particle size and the effect of an additional digestion time after the asphalt binder modification stage should also be evaluated.

## 5. Conclusions

BO can be considered a potential bio-modifier of the physical properties of asphalt binder because it shows positive effects on asphalt binder properties at high temperatures, such as: rotational viscosity, softening point and penetration. These properties can contribute to improve the rutting resistance of the asphalt pavement. In relation to low temperatures, the benefits of BO as a modifier are discrete, but contribute to increasing the viscoelastic range of asphalt binder.

## Figures and Tables

**Figure 1 materials-15-07000-f001:**
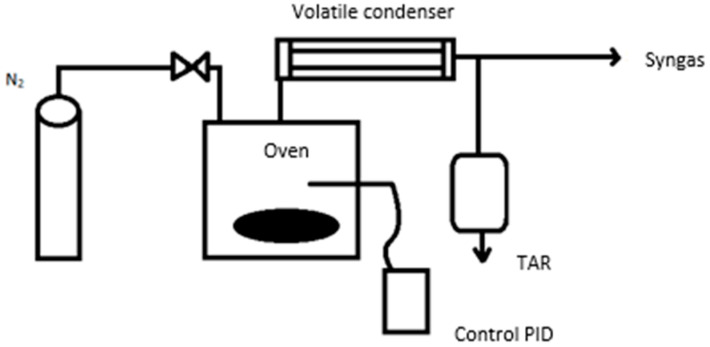
Schematic diagram of pyrolysis.

**Figure 2 materials-15-07000-f002:**
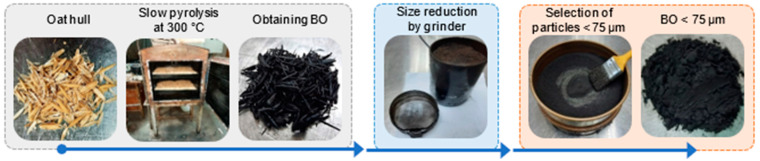
BO production sequence.

**Figure 3 materials-15-07000-f003:**
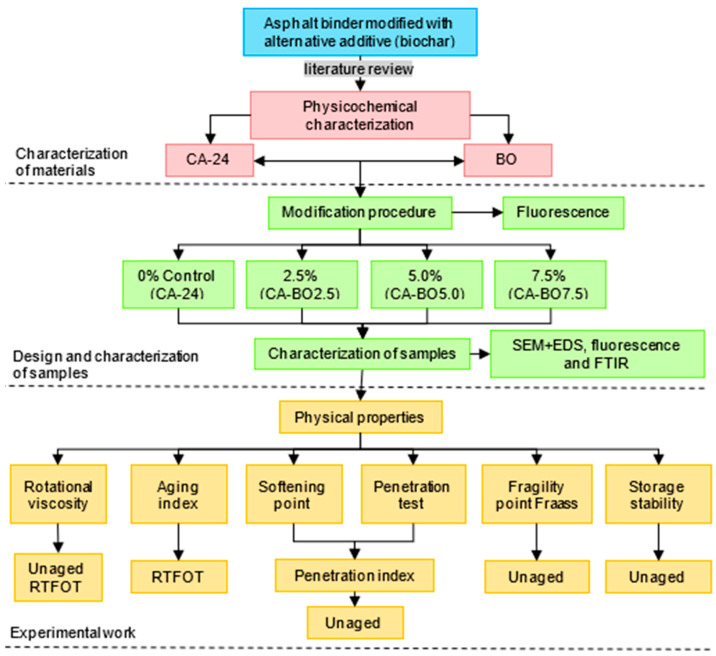
Experimental plan.

**Figure 4 materials-15-07000-f004:**
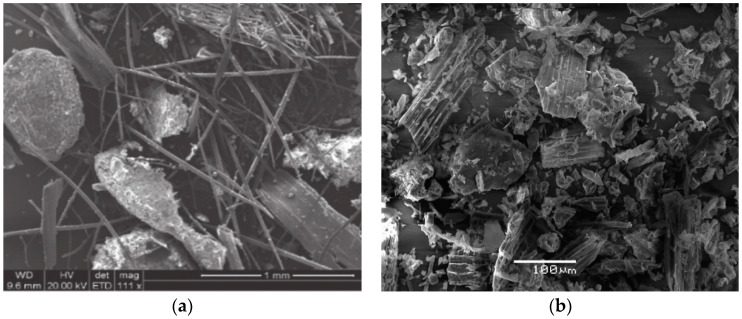
SEM micrograph: (**a**) Oat hulls and (**b**) BO.

**Figure 5 materials-15-07000-f005:**
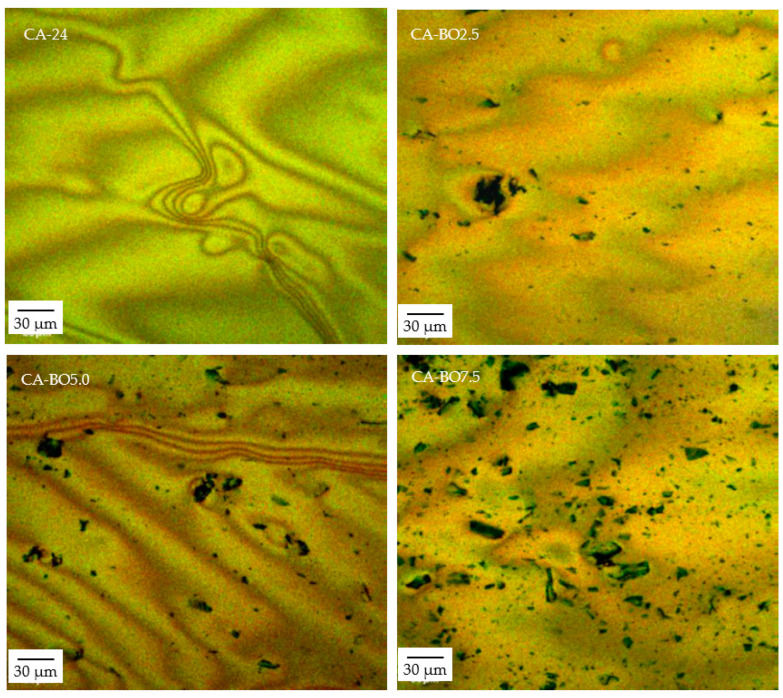
Fluorescence images of samples.

**Figure 6 materials-15-07000-f006:**
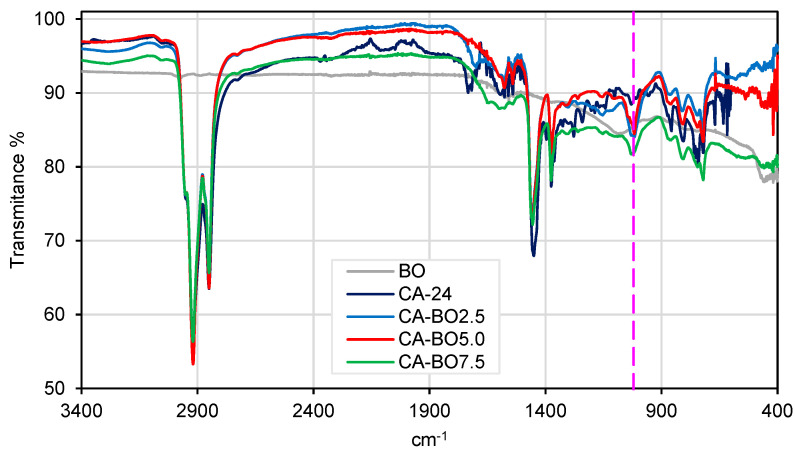
FTIR spectra of the samples analyzed.

**Figure 7 materials-15-07000-f007:**
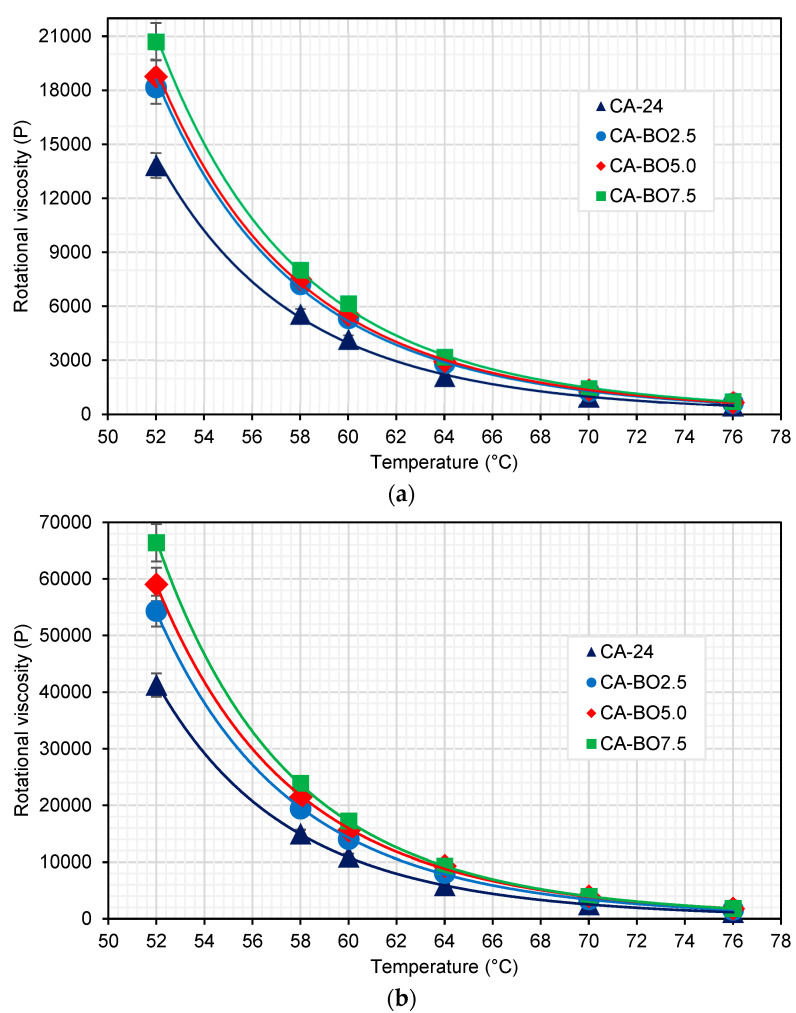
Rotational viscosity between 52 and 76 °C of asphalt binder samples analyzed at: (**a**) original state and (**b**) aged by RTFOT.

**Figure 8 materials-15-07000-f008:**
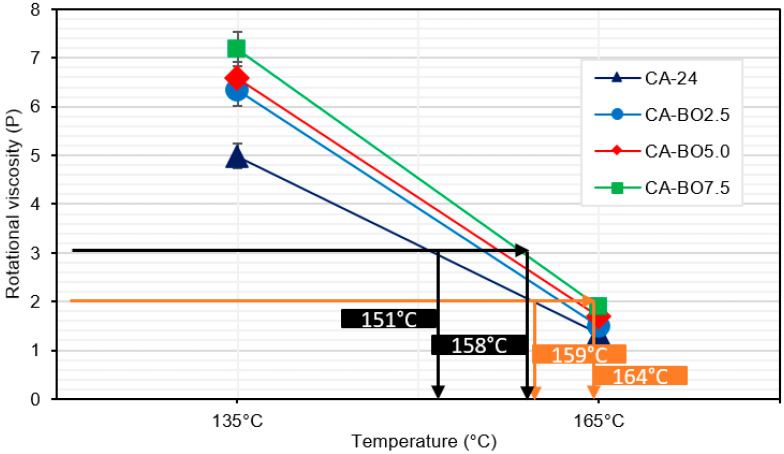
In black, mixing temperature (2 P) and in orange, compaction temperature (3 P) of the asphalt binder samples analyzed.

**Figure 9 materials-15-07000-f009:**
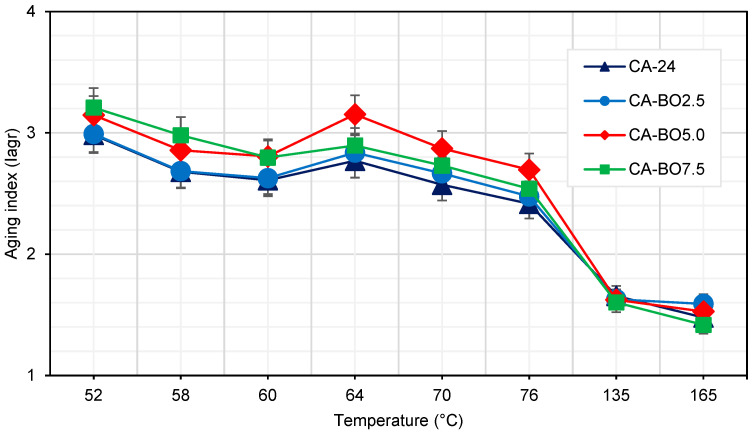
Aging index (I_ag_^r^) of the samples.

**Figure 10 materials-15-07000-f010:**
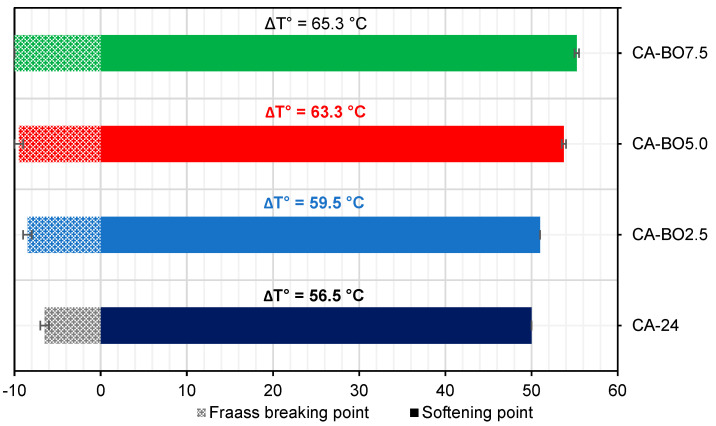
Viscoelastic range of the samples.

**Figure 11 materials-15-07000-f011:**
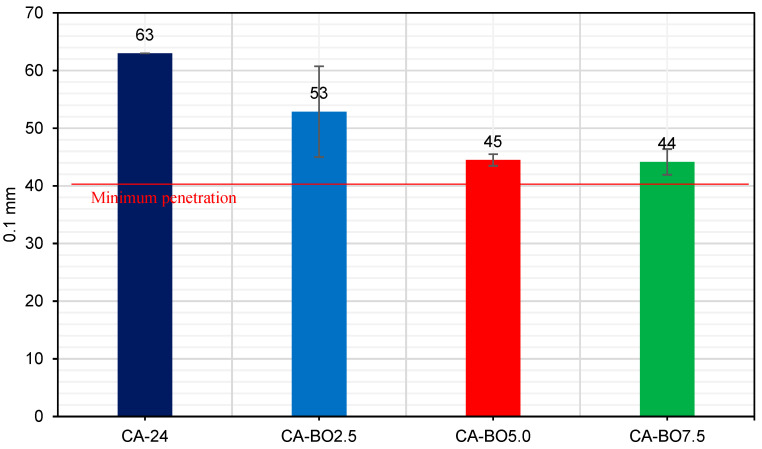
Penetration of the samples together with the required minimum penetration at 25 °C.

**Figure 12 materials-15-07000-f012:**
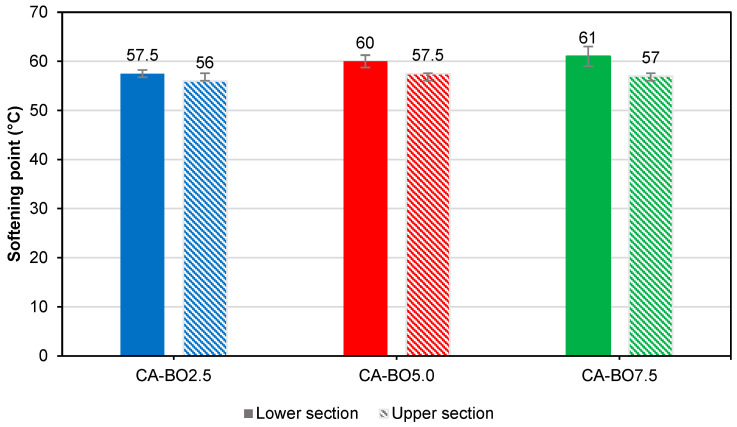
Storage stability of samples.

**Figure 13 materials-15-07000-f013:**
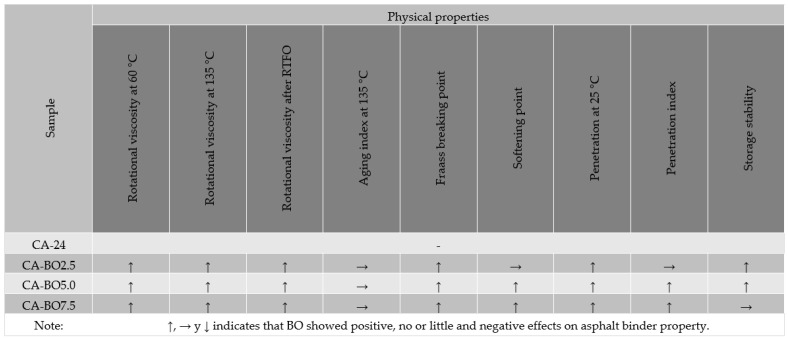
Summary of the effects of BO on the physical properties of asphalt binder.

**Table 1 materials-15-07000-t001:** Physical properties of reference asphalt binder (CA-24).

Tests	Specs. [32]	CA-24
Original viscosity at 60 °C (P)	Min. 2400	2940
Penetration at 25 °C, 100 g, 5 s (dmm)	Min. 40	63
Ductility at 25 °C, 5 cm/min (cm)	Min. 100	100
Trichloroethylene solubility (%)	Min. 99	99.8
Flash point (°C)	Min. 232	310
Softening point (°C)	-	52.2
Penetration index	−1.0 to + 1.0	−0.1
**RTFOT**		
Mass loss (%)	Max. 0.8	0.08
Viscosity at 60 °C (P)	-	7860
Ductility at 25 °C, 5 cm/min (cm)	Min. 100	100
Durability index	Max. 3.5	2.7

**Table 2 materials-15-07000-t002:** PI classification of asphalt binders [32].

IP Value	Description
PI > +1	Asphalts that are not very susceptible to temperature and show non-Newtonian flow behavior, with certain elasticity and thixotropy.
PI < −1	Asphalts that are highly susceptible to temperature and exhibit Newtonian flow behavior.
−1 < PI < +1	Asphalts that have rheological and flow characteristics intermediate between the two previous cases. Most of the asphalt binder used in paving has these characteristics.

**Table 3 materials-15-07000-t003:** Micro elemental analysis of the samples.

Chemical Element	BO		CA-24		CA-BO2.5		CA-BO5.0		CA-BO7.5	
	Content (%)	σ	Content (%)	σ	Content (%)	σ	Content (%)	σ	Content (%)	σ
Carbon (C)	70.51	3.09	92.60	3.93	91.39	3.13	86.54	8.39	87.03	5.10
Sulfur (S)	-	-	3.80	0.15	3.36	0.26	3.08	1.60	3.58	0.64
Oxygen (O)	22.72	0.48	2.92	1.69	3.82	2.95	8.24	6.73	5.16	4.01
Calcium (Ca)	1.38	0.29	-	-	0.67	0.39	-	-	-	-
Nitrogen (N)	-	-	7.87	4.54	-	-	4.31	2.49	7.35	4.24
Silicon (Si)	3.29	3.25	-	-	-	-	1.62	0.96	-	-
Potassium (K)	1.89	0.53	-	-	-	-	-	-	3.51	2.03
Phosphorus (P)	0.61	0.35	-	-	-	-	-	-	1.22	0.70

Note: σ: standard deviation.

**Table 4 materials-15-07000-t004:** Penetration index of the samples.

Sample	CA-24	CA-BO2.5	CA-BO5.0	CA-BO7.5
Penetration index	−0.7	−0.8	−0.6	−0.3

## Data Availability

Not applicable.

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
