# Peer review of "Effect of Biochar from Oat Hulls on the Physical Properties of Asphalt Binder"

_materials, 2022, doi:10.3390/ma15197000_

Round 1

Reviewer 1 Report

It is a good written paper. Author performed various tests to evaluate the effect of biochar on the various properties of asphalt binder. Some comments are:

1.      Page 4, figure 2 repeats.

2.      Section 2.3.2. What it he shear speed during the rotational viscosity test? Why did the author not perform long-term ageing (PAV) tests?

3.      Line 190. The detailed procedures of storage stability test should be described.

4.      The scales in Figure 4 (a) and (b) are different, making it difficult to compare.

5.       Table 3, is it possible to perform elemental analysis on BO sample? Why is the content of C, S and O of CA-BO5.0 not between the results of CA-BO2.5 and CA-BO7.5?

6.      Figure 6, what do the arrows point to?

7.      Line 373, the error message of reference.

8.      The discussion and conclusions should be written in one section.

Author Response

Dear reviewer, 

Reviewer 2 Report

The paper at this stage seems to be a case study rather than a scientific contribution. The theme of the research has certain practical significance, but the research lacks profundity and analytical precision. A thorough grammar check is needed. Hereby please find below some additional comments:

1. Write up of the manuscript: The overall writing of the manuscript is poor.

2. “asphlt binder” generally does not need abbreviations.

3. Abstract: No critical findings have been mentioned in the abstract section. “biochar from oat hulls is made up mainly of carbon(line20-21)” is common sense and should not be used as a main conclusion.

4. Introduction: line40-54 mentioned that the use of commercial polymer and nano-modifiers involves a considerable increase in the cost of the end product. If it is cost oriented, has the author considered the cost of biochar as a modifier? The advantages of biochar as a modifier are suggested to point out except reducing carbon emissions.

5. Figure 5: The legends are recommended to add in the figure for easy viewing.

6. Line 238: “cm-1”,please check! Line 239-240: The analysis is too simple, and relevant calculations are needed such as functional group index. Besides, the transmittance cannot reflect the absolute content, and a stable peak is usually used as a reference for relative content comparison.

7. Line 373: “¡Error! No se encuentra el origen de la referen-373 cia.”,line 383 “2 to 5 °Cs”, line 390 “[33], [34].”. Please check them!

8. The paper explains too much on the well-known knowledge and very little on the newer results and outcomes in Section 3. Rotational viscosity, Fraass breaking point, softening point, penetration, and storage stability are basic performance tests.

Author Response

Dear Reviewer,

Reviewer 3 Report

The manuscript studies the Effect of biochar from oat hulls on the physical properties of asphalt binder.

The manuscript includes an interesting work. The objective of the study is clear. The methods and the analyses are correct. This manuscript is properly, logically organized and it is innovative. I recommend accepting it after minor revision.

However, there are several points that need to be improved in order to match the quality standards of this Journal. Please consider the issues below:

1.       The introduction section should be expanded with a greater number of references. Please, explain why the use of oat hulls?

Are wastes of the author`s area? Is there a big generation of these wastes?

2.       How the authors control the parameters in the oven? Please explain.

3.       Have the authors any estimation about the economic or environmental benefits of this proposal?

4.       Figure 2 is repeated

5.       Explain in more detail the results of the physical test comparing with other bio-binder

Author Response

Dear Reviewer,
